# Fantasy: Transformer Meets Transformer in Text-to-Image Generation

## Abstract

We present Fantasy, an efficient text-to-image generation model marrying the decoder-only Large Language Models (LLMs) and transformer-based masked image modeling (MIM). While diffusion models are currently in a leading position in this task, we demonstrate that with appropriate training strategies and high-quality data, MIM can also achieve comparable performance. By incorporating pre-trained decoder-only LLMs as the text encoder, we observe a significant improvement in text fidelity compared to the widely used CLIP text encoder, enhancing the text-image alignment. Our training approach involves two stages: 1) large-scale concept alignment pre-training, and 2) fine-tuning with high-quality instruction-image data. Evaluations on FID, HPSv2 benchmarks, and human feedback demonstrate the competitive performance of Fantasy against state-of-the-art diffusion and autoregressive models.

## 1 Introduction

Recent advances in text-to-image (T2I) models [3, 5, 12] have become focal points within the computer vision field. Most advances in T2I models, focused on generating high-quality images based on relatively short descriptions, struggle with intricate long-text semantic alignment due to inherent structure constraints and data limitations. Text encoders used for T2I fall into three categories: CLIP [30], encoder-decoder LLMs, and decoder-only LLMs. Models using encoder-decoder LLMs like T5-XXL [31] have shown improved text-image alignment over CLIP by exploiting enhanced text understanding, increasing token capacity, yet without delving into the semantic alignment for longer texts. ParaDiffusion [43] indicates that directly aligning text embeddings with visual features without prior image-text knowledge is not the most effective approach. Previous works [38, 45] have highlighted shortcomings in existing text-image datasets [37], including image-text mismatches, a lack of informative content, and a pronounced long-tail effect. These deficiencies notably impair training efficiency for T2I models and restrict their ability to learn complex semantic alignment.

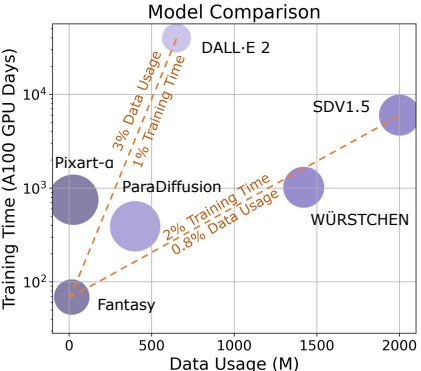

Figure 1: Comparison of data usage, training time and image quality. Colors from dark to light represent parameters increasing in size, and circles from small to large indicate improvements in image quality.

Existing diffusion-based T2I models [33, 5, 9, 26] have achieved unprecedented quality. However, as detailed in Fig. 1, these advanced models come with significant computational demands. The

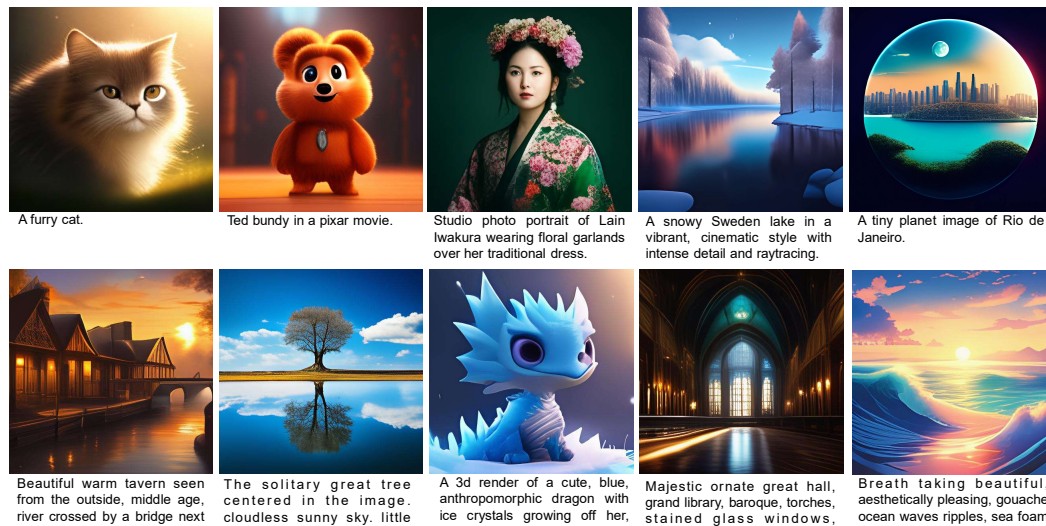

| | | | | |
|---|---|---|---|---|
| A furry cat. | Ted bundy in a pixar movie. | Studio photo portrait of Lain Iwakura wearing floral garlands over her traditional dress. | A snowy Sweden lake in a vibrant, cinematic style with intense detail and raytracing. | A tiny planet image of Rio de Janeiro. |
| Beautiful warm tavern seen from the outside, middle age, river crossed by a bridge next to the tavern, crepuscular light. | The solitary great tree centered in the image. cloudless sunny sky. little islands in the flooded plain. | A 3d render of a cute, blue, anthropomorphic dragon with ice crystals growing off her, sharp focus. | Majestic ornate great hall, grand library, baroque, torches, stained glass windows, moonlight rays, dreamy mood. | Breath taking beautiful, aesthetically pleasing, gouache ocean waves ripples, sea foam, sunset, digital concept art. |

Figure 2: Samples produced by Fantasy ($512 \times 512$). Each image, generated in 1.26 seconds (without super-resolution models), is accompanied by a descriptive caption showcasing diverse styles and comprehension.

considerable expenses of these models create significant barriers for researchers and entrepreneurs. Meanwhile, economical text-to-image models [25, 15, 48] compromise on image quality, yielding lower resolution and diminished aesthetic appeal.

Given these challenges, a pivotal question arises: *Can we develop a **resource-efficient**, **high-quality** image generator for **long** instructions?* In this paper, we present Fantasy, significantly reducing training demands while maintaining the capability of instruction understanding and competitive image generation quality, as shown in Fig. 2. To achieve this, we propose three core designs:

**Efficient T2I netwrok.** To leverage the powerful understanding ability of a decoder-only LLM, we choose the lightweight Phi-2 [24] as our text encoder. We derive discrete image tokens from a pre-trained VQGAN [27], and employ Transformer-based masked image modeling (MIM) as our T2I architecture. We also utilize the pre-trained VQGAN decoder [27] for pixel space restoration.

**Hierarchical Training strategy.** We propose a thoughtfully two-stage training strategy to address the high computational demands of current leading models while maintaining competitive performance: (1) large-scale concept alignment pre-training, (2) high-quality instruction-image fine-tuning. To facilitate a coarse image-text alignment, we initially train the T2I model from scratch using relatively lower-quality data. We then fine-tune the pre-trained T2I model and LLM on text-image pair data rich in information density with superior aesthetic quality.

**High-quality data.** To achieve rough alignment while pre-training, we select the large-scale dataset LAION-2B [37] and employ the filtering strategy proposed by DataComp [14]. We collect long-text prompts and corresponding high-quality synthesized images for instruction tuning, including DiffusionDB [42] and JourneyDB [39]. We further filter and discard texts with special characters and data containing violence or pornography, retaining only instructions exceeding 30 words.

Our main contributions are summarized as follows:

1. We present Fantasy, a novel framework that is the first to integrate a lightweight decoder-only LLM and a Transformer-based MIM for text-to-image synthesis, allowing for long-form text alignment.

2. We show that our two-stage training strategy with high-quality data enables MIM to achieve comparable performance at a significantly reduced training cost.

3. We provide comprehensive validation of the model's efficacy based on automated metrics and human feedback for visual appeal and text faithfulness.

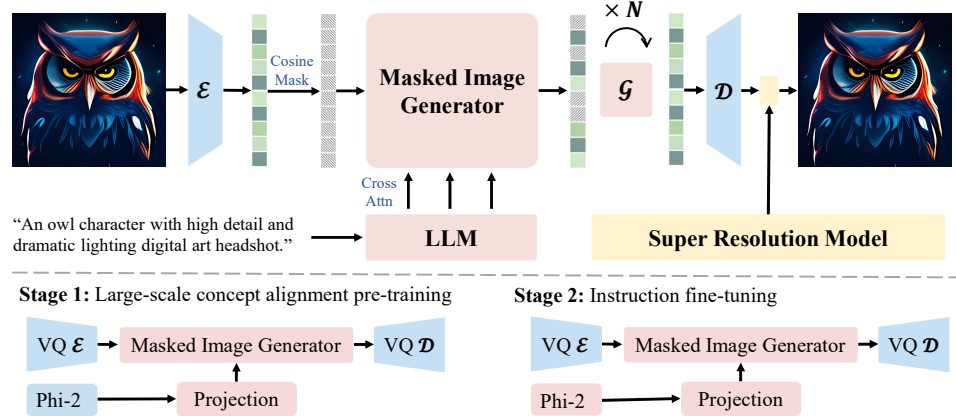

Figure 3: **(Up)** Overview of Fantasy featuring text encoder, VQGAN (encoder $\mathcal{E}$ and decoder $\mathcal{D}$), masked image generator $\mathcal{G}$, and super-resolution model. **(Down)** Our training pipeline involves two stages. The red parts are trainable and the blue parts are frozen; the yellow part is optionally utilized during inference.

## 2 Method

### 2.1 Problem Formulation

As depicted in Fig. 3, Fantasy consists of a pre-trained text encoder $\mathcal{T}$, a transformer-based masked image generator $\mathcal{G}$, a sampler $\mathcal{S}$, a frozen VQGAN, and a pre-trained super-resolution model. $\mathcal{T}$ maps a text prompt $t$ to a continuous embedding space. $\mathcal{G}$ processes a text embedding $e$ to generate logits $l$ for the visual token sequence. $\mathcal{S}$ draws a sequence of visual tokens $v$ from logits via iterative decoding [4], which runs $N$ steps of inference conditioned on the text embeddings $e$ and visual tokens decoded from previous steps. Finally, $\mathcal{D}$ maps the sequence of discrete tokens to pixel space $Z$. To summarize, given a text prompt $t$, an image $\hat{x}$ is synthesized as follows:

$$\hat{x} = \mathcal{D}(\mathcal{S}(\mathcal{G}, \mathcal{T}(t))), \quad l_n = \mathcal{G}(v_n, \mathcal{T}(t)), \quad v_n = \mathcal{M}(\mathcal{E}(x)) \tag{1}$$

where $n$ is the synthesis step, and $l_n$ are logits, from which the next set of visual tokens $v_{n+1}$ are sampled. $\mathcal{M}$ denotes the masking operator that applies masks to the token in $v_n$. We refer to [4, 3] for details on the iterative decoding process. The Phi-2 [24] for $\mathcal{T}$ and VQGAN [8] for encoder $\mathcal{E}$ and decoder $\mathcal{D}$ are used. $\mathcal{G}$ is trained on a large text-image pairs $D$ using masked visual token modeling loss:

$$\mathcal{L} = \mathbb{E}_{(x,t) \sim D}\left[CE\left(l_N, \mathcal{E}(x)\right)\right], \tag{2}$$

where $CE$ is a weighted cross-entropy calculated by summing only over the unmasked tokens.

### 2.2 Model Architecture

#### 2.2.1 VQGAN as Image Processor

VQGAN [8] is capable of transforming each image into discrete tokens with higher-level semantic information from a learned codebook, while ignoring low level noise. The autoregressive tokens prediction of VQGAN shares the same form as text tokens generated by LLMs. Prior research [46] has shown that unifying vision and language by the same token space could enhance the coherency for vision-text alignment. Furthermore, compared with RGB pixels, the visual token representation has proven to reduce disk storage and improve the capability of robustness and generalization.

To reduce the computational burden, we initially compress an RGB image $v \in \mathbb{R}^{H \times W \times 3}$ into a diminished representation with a resolution of $h \times w \times 3$, where $h = H/f$ and $w = W/f$, with $f$ denoting the downsampling factor. We then employ a pre-trained $f16$ VQGAN [27] encoder $\mathcal{E}$ to quantizate images $x \in \mathbb{R}^{3 \times 256 \times 256}$ into discrete tokens of spatial dimensions $16 \times 16$ from a pre-trained codebook $\mathcal{Z} = \{z_k\}_{k=1}^K$ consisting of $K = 8192$ vectors, resulting in the quantized representation $z = \mathcal{E}(x, \mathcal{Z})$.

### 2.2.2 LLM as Text Encoder

Recent studies [10, 5, 3] tend to use encoder-decoder LLMs [31] for text encoding over CLIP [30], which is adept at handling tasks that involve complex mappings between input and output sequences. Due to the tremendous success of ChatGPT, attention has been drawn to models that consist solely of a decoder. Also, [43] presents an insight that efficiently fine-tuning a more powerful decoder-only LLM can yield stronger performance in long-text alignment. Consequently, to capitalize on the enhanced semantic comprehension and generalization potential of LLMs while simultaneously reducing the training burden, we employ Phi-2 [24], a state-of-the-art, lightweight LLM, as the text encoder.

Given the text prompt $t$, Fantasy first passes it through Phi-2, extracting the text embedding from the last hidden layer $L$. However, typically, decoder-only architectures are not adept at feature extraction and mapping tasks. [23] proposes that the conceptual representations learned by LLM's are roughly linearly mappable to those learned by models trained on vision tasks. Therefore, the embedding vectors are linearly projected to the hidden size of the image generator $\mathcal{G}$:

$$c = \mathcal{P}(\mathcal{T}_L(t)) \tag{3}$$

where $\mathcal{T}(\cdot)$ denotes the decoder-only Phi-2 and $L$ is the index of the last hidden layer. $\mathcal{P}$ represents the projection from text space to visual space, and $c$ is the text feature suitable for the image generator.

### 2.2.3 MIM as Image Generator

MIM narrows the gap between its modeling and the extensively studied area of language modeling, making it straightforward to leverage the findings of the LLMs research community. Therefore, we adopt a masked transformer as the image generator backbone of Fantasy [46].

During training, we leave the projected text embeddings $c$ unmasked and the image tokens $z$ are masked at a variable masking rate based on a Cosine scheduling $\mathcal{M}$ as [4, 3]. Specifically, for each training example, we sample a masking rate $r$ from $[0, 1]$ from a truncated $arccos$ distribution with density function $p(r) = \frac{2}{\pi}(1 - r^2)^{-\frac{1}{2}}$. While autoregressive methods learn fixed-order token distributions $P(z_i|z_{<i})$, random masking with variable ratios enables learning $P(z_i|z_{\neq i})$ for any token subset, crucial for our parallel sampling scheme. The sampling of a new state $s_{n+1}$ at each successive step is conditioned on the previous state and the specified text condition $c$:

$$P(s \mid c) = \int P(s_N \mid s_{N-1}, c) \prod_{n=1}^{N-1} P(s_n \mid s_{n-1}, c) \, ds_1 \ldots ds_{N-1} \tag{4}$$

For each training example, the most confidently predicted tokens are revealed at each step $n$, maintaining $\cos\left(\frac{n}{N} \cdot \frac{\pi}{2}\right)$ masked until reaching $N$ total steps.

For the base model, we use a variant of MaskGiT [4], a masked image generative Transformer to predict randomly masked tokens by attending to tokens in all directions. Leveraging the multi-layered structure of the Transformer, we have developed scalable image generators with varying layer counts, ranging in size from 257M parameters to 611M parameters (for the image generator; the Phi-2 model has an additional 2.7B parameters). We first employ a series of Cross Attention blocks to optimize text-driven feature extraction, before passing through $O$ layers of the masked image generator. Each layer $o$ of the Transformer is again formed by Multi-Head Self-Attentuib(MSA), LayerNorm (LN), Cross Attention (CA) and Multi-Layer Perceptron (MLP) blocks:

$$Y_o = \text{MSA}(\text{LN}(Z_o)), \quad Z_{o+1} = \text{MLP}(\text{CA}((\text{LN}(Y_o), c))). \tag{5}$$

At the output layer, to reduce the training burden, ConvMLP [18] is utilized to transform masked image embeddings into logits sets, aligning with the VQGAN codebook dimensions. Eventually, the reconstructed lower-resolution tokens are restored with the pre-trained $256 \times 256$ resolution VQGAN decoder to the pixel space, resulting in the generated image $\hat{x}$:

$$\hat{x} = \mathcal{D}(\text{ConvMLP}(Z_O), \mathcal{Z}) \tag{6}$$

### 2.3 Training Strategy

Fig. 3 illustrates Fantasy's two-stage training approach. Following prior works[43, 35, 9], we employ large-scale pre-training to achieve general text-image concept alignment, and simultaneous fine-tuning of Phi-2 [24] and the masked image generator using high-quality instruction-image pairs.

**Pre-training Stage.** To perform general text-image concept alignment, the VQGAN and LLM weights are frozen, and only the image generator is pre-trained on deduplicated LAION-2B [37] with images above a 4.5 aesthetic score. We exclusively preserve prompts in English, filter out images above a 50% watermark probability or above a 45% NSFW probability, yielding a final set of 9 million images. Since the computational cost of upsampling is much lower than training a super-resolution model, Fantasy is started with training at a resolution of $256 \times 256$. Note that the pre-training only needs approximate image-text alignment, substantially lowering the training costs.

**Fine-tuning Stage.** [43] has proven that LLMs trained solely on text data lack prior image-text knowledge, and that merely aligning their text embeddings with visual features might not be optimal. Therefore, in the second stage, we gather an internal dataset of 7 million high-quality instruction-image pairs to fine-tune both the Phi-2 model and the image generator of Fantasy, which ensures enhanced compatibility of text embeddings within the text-image pair space, facilitating the use of decoder-only LLMs in text-to-image generation tasks and harnessing their inherent advantages. To prevent catastrophic forgetting in LLMs and preserve their understanding abilities during training, we select questions from BIG-bench [2] and monitor the common sense question-answering ability of Phi-2 in real-time throughout the training process. We construct our training dataset for the fine-tuning stage by incorporating JourneyDB [39] and an internal synthetic dataset to enhance the aesthetic quality of generated images beyond realistic photographs. To facilitate instruction-image alignment learning, we retain only data with descriptions exceeding 30 words, as these provide enough detailed insights into the image objects, including attributes and spatial relations.

With this approach, Fantasy trains a 0.6B parameter T2I model in about 69 A100 GPU days, significantly reducing computation compared to existing diffusion-based methods, while maintaining comparable visual and numerical fidelity. Throughout this paper, we present a comprehensive evaluation of Fantasy's efficacy, showcasing the potential in training high-quality transformer-based image synthesis models compared to diffusion-based models in future.

## 2.4 High-quality Data Collection

To ensure rough alignment in the pre-training phase, we utilize the large-scale dataset LAION-2B [37] and apply the filtering strategy developed by DataComp [14]. Furthermore, we gather long-text prompts and corresponding high-quality images to achieve finer-grained text-image alignment through instruction tuning. CapsFusion [47] employs a fine-tuned LLaMA [40] for recaptioning LAION-2B [37] and LAION-COCO [1]. However, this approach still results in suboptimal image quality and occasional mismatches between images and text. SAM-LLAVA [5] utilizes LLaVA [20] to recaption the SAM dataset [17], which leads to images with blurred faces, a consequence of the dataset's inherent face-blurring. Therefore, we shift focus to synthesize images, mainly including DiffusionDB [42] and JourneyDB [39], produced by Stable Diffusion and MidJourney, respectively. To augment the diversity of the images, we minimize the use of datasets from specific domains, such as gaming and anime. Furthermore, we implement filtering to discard texts with special characters and data containing violence or pornography, retaining only instructions exceeding 30 words.

## 3 Experiments

In this section, we outline detailed training, inference, and evaluation protocols, followed by comprehensive comparisons across three key metrics.

### 3.1 Implementation Details

**Training Details.** Different from the prior works [9, 43, 32, 34], we used a lightweight but powerful decoder-only large language model Phi-2 [24] as the text encoder. Diverging from prior approaches that extract a standard and fixed short text tokens, we extend the extraction to 256 tokens to master long-term instruction-image alignment, ensuring precise alignment for more fine-grained prompts. For the entire training process, we train Fantasy on $4 \times$A100 80G GPUs and set the accumulation step to 2. At different stages, we employ varying learning rate strategies with single-cycle cosine annealing decay. Furthermore, the AdamW optimizer [22] is utilized with a weight decay of 0.01. Fantasy trains a 0.6B parameter T2I model in about 84.5 A100 GPU days, significantly reducing computation compared to existing diffusion-based methods as shown in Fig. 1.

Table 1: Evaluation of diffusion (upper) and transformer (down) models on HPSv2. We underline the highest value and color the first above Fantasy in blue .

| Model | Type | Params | Animation | Concept-art | Painting | Photo | DrawBench [36] |
|-------|------|--------|-----------|-------------|----------|-------|----------------|
| GLIDE [25] | Diff | 5.0B | $23.34 \pm 0.198$ | $23.08 \pm 0.174$ | $23.27 \pm 0.178$ | $24.50 \pm 0.290$ | $25.05 \pm 0.84$ |
| VQ-Diffusion [15] | Diff | 0.37B | $24.97 \pm 0.186$ | $24.70 \pm 0.149$ | $25.01 \pm 0.145$ | $25.71 \pm 0.222$ | $25.44 \pm 0.83$ |
| Latent Diffusion [34] | Diff | 1.45B | $25.73 \pm 0.125$ | $25.15 \pm 0.140$ | $25.25 \pm 0.178$ | $26.97 \pm 0.183$ | $26.17 \pm 0.85$ |
| DALL·E 2 [26] | Diff | 6.5B | $27.34 \pm 0.175$ | $26.54 \pm 0.127$ | $26.68 \pm 0.156$ | $27.24 \pm 0.198$ | $27.16 \pm 0.64$ |
| Stable Diffusion v1.4 [33] | Diff | 0.8B | $27.26 \pm 0.156$ | $26.61 \pm 0.082$ | $26.66 \pm 0.143$ | $27.27 \pm 0.226$ | $27.23 \pm 0.57$ |
| Stable Diffusion v2.0 [33] | Diff | 0.8B | $27.48 \pm 0.174$ | $26.89 \pm 0.076$ | $26.86 \pm 0.120$ | $27.46 \pm 0.198$ | $27.31 \pm 0.68$ |
| DeepFloyd-XL [11] | Diff | 4.3B | $27.64 \pm 0.108$ | $26.83 \pm 0.137$ | $26.86 \pm 0.131$ | $27.75 \pm 0.171$ | $27.64 \pm 0.72$ |
| LAFITE [48] | Trans | 0.075B | $24.63 \pm 0.101$ | $24.38 \pm 0.087$ | $24.43 \pm 0.155$ | $25.81 \pm 0.213$ | $25.23 \pm 0.72$ |
| FuseDream [21] | Trans | - | $25.26 \pm 0.125$ | $25.15 \pm 0.107$ | $25.13 \pm 0.183$ | $25.57 \pm 0.248$ | $25.72 \pm 0.71$ |
| DALL·E mini [7] | Trans | 0.4B | $26.10 \pm 0.132$ | $25.56 \pm 0.137$ | $25.56 \pm 0.112$ | $26.12 \pm 0.233$ | $26.34 \pm 0.76$ |
| VQGAN + CLIP [8] | Trans | 0.2B | $26.44 \pm 0.152$ | $26.53 \pm 0.075$ | $26.47 \pm 0.111$ | $26.12 \pm 0.210$ | $26.38 \pm 0.43$ |
| CogView2 [12] | Trans | 6B | $26.50 \pm 0.129$ | $26.59 \pm 0.119$ | $26.33 \pm 0.100$ | $26.44 \pm 0.271$ | $26.17 \pm 0.74$ |
| Fantasy (ours) | Trans | 0.6B | **$27.03 \pm 0.131$** | **$26.66 \pm 0.117$** | **$26.72 \pm 0.176$** | **$26.80 \pm 0.174$** | **$26.78 \pm 0.523$** |

Table 2: Comparison with recent T2I models. 'Trained' indicates the model develops a text encoder from scratch, foregoing a pre-trained one.

| Method | Type | Text Encoder | #Params | #Images | FID-30K ($\downarrow$) |
|--------|------|--------------|---------|---------|------------------------|
| LDM [34] | Diff | Trained | 1.4B | 400M | 12.64 |
| GLIDE [25] | Diff | Trained | 5.0B | - | 12.24 |
| DALL·E 2 [26] | Diff | CLIP | 6.5B | 650M | 10.39 |
| Stable Diffusion v1.5 [33] | Diff | CLIP | 0.9B | 2000M | 9.62 |
| SD XL [29] | Diff | CLIP | 2.6B | - | >18 |
| Würstchen [28] | Diff | CLIP | 0.99B | 1420M | 23.6 |
| ParaDiffusion [43] | Diff | LLaMA V2 | 1.3B | >300M | 9.64 |
| Pixart-$\alpha$ [5] | Diff | T5 | 0.6B | - | 5.51 |
| Cogview2 [12] | Trans | CogLM | 6B | 35M | 24.0 |
| Muse [3] | Trans | T5-XXL | 3B | 460M | 7.88 |
| Fantasy | Trans | Phi-2 | 0.6B | 16M | 23.4 |

**Inference Details.** We use $N = 32$ sampling steps in all of our evaluation experiments. Since Fantasy is trained at a resolution of $256 \times 256$, we employ the pre-trained diffusion-based super-resolution model StableSR [41] to upscale images to $512 \times 512$.

**Evaluation Metrics.** We comprehensively evaluate Fantasy via four primary metrics, i.e., alignment on HPSv2 [44], FID [16] on MSCOCO dataset [19] and human evaluation on a collected dataset.

## 3.2 Performance Comparisons and Analysis

**Results on HPSv2.** We utilize HPSv2 [44] as our primary automated metric, a preference prediction model which can be used to compare images generated with the same prompt across five categories: anime, concept art, paintings, photography, and DrawBench [36]. We present the results of HPSv2 between Fantasy and other state-of-the-art generative models in Tab. 1. Fantasy exhibited outstanding performance across all key aspects among previous Transformer-based methods like CogView2 [12], which is expected. The results also reveal its competitive performance compared to prior diffusion-based methods, especially in concept-art and painting, demonstrating similar performance to DALL·E 2 [26]. This remarkable performance is primarily attributed to the text-image alignment learning in fine-tuning stage, where high-quality text-image pairs were leveraged to achieve superior alignment capabilities. In comparison, DeepFloyd-XL and other diffusion-based models achieve better scores, while utilizing larger models with significantly higher compute budget.

**Results on FID.** We employ FID [16] to evaluate our models on COCO-30K [19]. To allow for a fair comparison, all images are downsampled to $256 \times 256$ pixels. The comparison between our method and other methods in FID, and their training time is summarized in Tab. 2. We observe that the FID of Fantasy is substantially higher compared to other state-of-the-art models. Visual inspections reveal that images generated by Fantasy are smoother than those from other leading T2I models. This discrepancy is most noticeable in real-world images like COCO, on which we compute the FID-metric. Although the state-of-the-art models [43, 11, 29] exhibit lower FID, it relies on unaffordable resources. Furthermore, prior studies [29, 5, 11] have demonstrated that FID may not

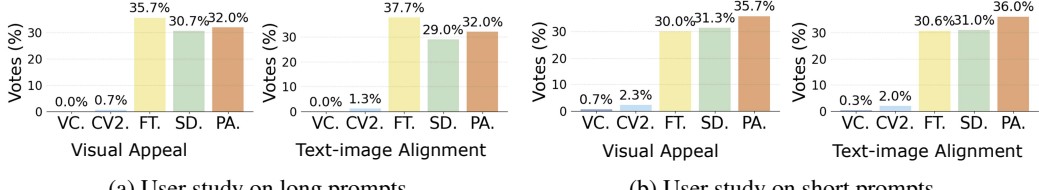

(a) User study on long prompts.      (b) User study on short prompts.

Figure 4: User study on prompts with different length. VC. , CV2. , FT. , SD. , and PA. refer to VQGAN+CLIP [8], CogView2 [12], our Fantasy, Stable Diffusion v2.0 [33], and Pixart-$\alpha$ [5].

be an appropriate metric for image quality evaluation, as a lower score does not necessarily reflect superior image generation, and it is more authoritative to use the evaluation of human users.

## 3.3 Results on Human Evaluation

Following prior works [5, 43, 28], we also conduct a study with human participants to supplement our evaluation and provide a more intuitive assessment of Fantasy's performance. Participants are asked to select a preference of the images based on the visual appeal of the generated images and the precision of alignments between the text prompts and the corresponding images.

As involving human evaluators can be time-consuming, we choose the top-performing open-source diffusion-based models (e.g., SD XL [33], and Pixart-$\alpha$ [5]) and transformer-based models (e.g., VQGAN+CLIP [8] and CogView2 [12]) as our baseline, which are accessible through APIs and capable of generating images. We randomly select a total of 600 prompts from existing prompt sets (e.g., ParaPrompt [43], ViLG-300 [13], COCO Captions [6]). To comprehensively contrast the capabilities of Fantasy and other models in interpreting text prompts of varying lengths, we allocate one subset to consist of 300 prompts ranging from 10 to 30 characters and another subset comprising 300 prompts exceeding 30 characters. For each model, we use a consistent set to generate images, which are then evaluated by 50 individuals.

Fig. 4a clearly demonstrates that images generated on relatively long text prompts (longer than 30 words) by Fantasy are distinctly favored among the four models in both two perspective, especially for text-image alignment, aligning closely with the intended use case of Fantasy. As illustrated in Fig. 4b, for text prompts shorter than 30 words, our model outperforms existing open-source Transformer-based models in fidelity and alignment for shorter prompts. Our model slightly lags behind diffusion-based models in visual appeal, limited by the 8,192 size of VQGAN's codebook and not targeting visual appeal. Simultaneously, Fantasy lacks a distinct advantage in text-image alignment in the short subset. We hypothesize that this is due to two main reasons: diffusion-based models' ability to handle shorter prompts, and vague prompts generating diverse images that make preferences more subjective, thus biasing outcomes towards aesthetically superior images. In summary, the human preference experiments confirm the observation made in the HPSv2 benchmarks.

## 3.4 Case Study

Fig. 5 vividly illustrates Fantasy's superior visual appeal and text-image alignment over leading open-source transformer-based T2I models [12, 8] and diffusion-based T2I models [29, 26]. Fantasy significantly surpasses existing transformer-based T2I models, matches the performance of SDXL [29], and qualitatively outperforms Dall·E 2 [26]. Despite being trained on images with a resolution of $256 \times 256$, Fantasy ensures generated low-resolution images contain sufficient details, indirectly supporting long prompts. Limited by computing resources, we haven't

A close-up photo of a person. The subject is a male. He was wearing a wide-brimmed hat, a gray-white beard on his face, a brown coat. His facial expression looked pensive and serious, with the clear blue sky in the background.

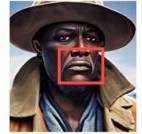

**ParaDiffusion**      **Fantasy**

A young man wearing a black leather jacket and tie stood behind an old door, his gaze firmly fixed on the camera. The door had patterns of leaves and flowers on it, revealing a yellow background. His hair was casually curled and he appeared to be deep in thought or contemplating something.

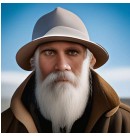

**ParaDiffusion**      **Fantasy**

Figure 6: Visual Comparison with ParaDiffusion [43]: Red markings and boxes highlight text misalignments in images generated by ParaDiffusion.

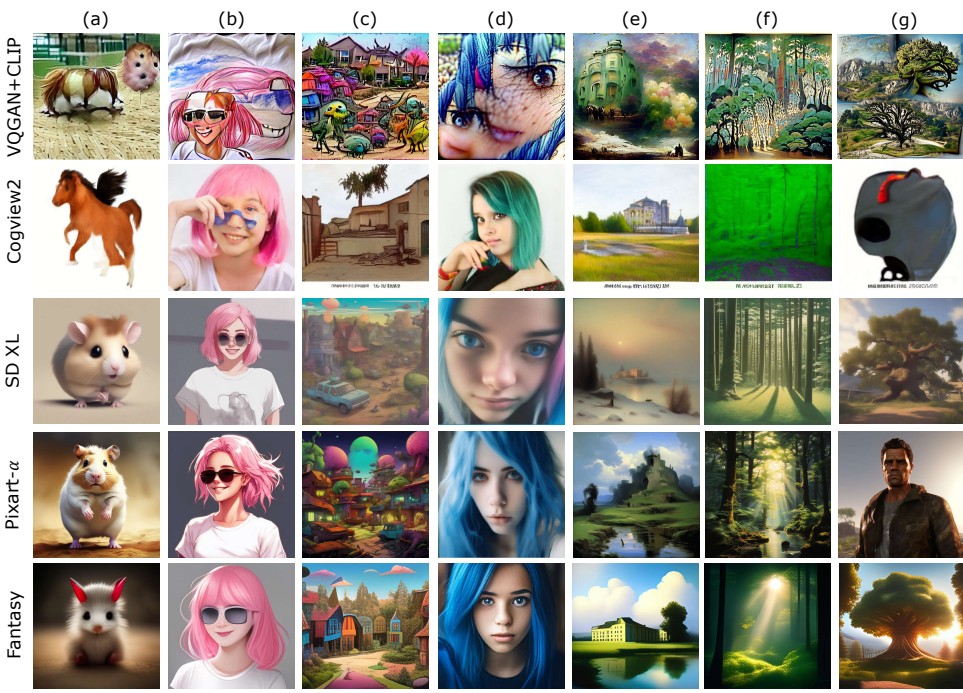

Figure 5: Visual comparison with existing T2I models. (a) *A hamster resembling a horse.* (b) *A frontal portrait of a anime girl with chin length pink hair wearing sunglasses and a white T-shirt smiling.* (c) *A colorful illustration of a suburban neighborhood on an ancient post-apocalyptic planet featuring creatures made by Jim Henson's workshop.* (d) *A blue-haired girl with soft features stares directly at the camera in an extreme close-up Instagram picture.* (e) *A building in a landscape by Ivan Aivazovsky.* (f) *Aoshima's masterpiece depicts a forest illuminated by morning light.* (g) *The image is a highly detailed portrait of an oak in GTA V, created using Unreal Engine and featuring fantasy artwork by various artists.*

Table 3: Ablation study on two stages with the best bolded. 'Base' indicates the model after the pre-training stage.

| Model | Training Part | Animation | Concept-art | Painting | Photo | DrawBench [36] |
|---|---|---|---|---|---|---|
| Base | MIM | $25.27 \pm 0.190$ | $24.20 \pm 0.166$ | $24.60 \pm 0.146$ | $25.32 \pm 0.208$ | $25.49 \pm 0.230$ |
| Fantasy | MIM+Phi-2 | **27.03±0.131** | **26.66±0.117** | **26.72±0.176** | **26.80±0.174** | **26.78±0.521** |

trained on higher resolutions like $512 \times 512$ but aim to enhance Fantasy by training at higher resolutions in the future.

ParaDiffusion [43] pioneers the use of decoder-only large language models as text encoders in text-to-image generation. As illustrated in Fig. 6, our observations suggest that Fantasy more closely aligns details with prompts than ParaDiffusion [43].

## 4 Ablation Study

This section analyzes the effects of LLMs fine-tuning, and model scale on Fantasy's performance through ablation studies. More ablation study refers to appendix.

### 4.1 Effect of Language Model Fine-tuning

To assess the effect of training strategies on the comprehension of complex instructions, we perform a human preference evaluation, as detailed in Sec. 3.3, using a subset of 300 prompts longer than 30 characters. 'Base' denotes general text-image alignment with filtered LAION-2B [1] in the pre-training stage. Compared to the base model, our synergy fine-tuning with Phi-2 demonstrates a notable improvement in all aspects in Tab. 3.

Table 4: Ablation study on models at different scales with the best **bolded**. DB. represents DrawBench [36].

| Layers | Param | Animation | Concept-art | Painting | Photo | DB. |
|--------|-------|-----------|-------------|----------|-------|-----|
| 6 | 257M | 25.79±0.15 | 25.84±0.11 | 25.92±0.19 | 25.63±0.18 | 25.18±0.22 |
| 12 | 421M | 26.34±0.17 | 26.29±0.06 | 26.45±0.17 | 26.19±0.17 | 25.68±0.14 |
| 22 | 611M | **27.03±0.13** | **26.66±0.11** | **26.72±0.17** | **26.80±0.17** | **26.78±0.52** |

Table 5: Training cost for Fantasy at 3 different scales. BS. denotes batch size and LR. denotes learning rate.

| Layers | Pre-training | | | Fine-tuning | | |
|--------|------------|-----|-----|------------|-----|-----|
| | Steps (K) | BS. | LR. | Steps (K) | BS. | LR. |
| 6 | 180 | 768 | 1e-4 | 180 | 192 | 1e-4 |
| 12 | 220 | 768 | 1e-4 | 250 | 192 | 1e-4 |
| 22 | 370 | 256 | 5e-4 | 280 | 128 | 3e-4 |

## 4.2 Scale of Image Generator

The hierarchical structure of the Transformer allows us to train image generators with varying numbers of Transformer layers. As shown in Tab. 4, we evaluate models of different sizes on the HPSv2 benchmark. The insight indicates that as trainable parameters increase from 257 million to 611 million, performance consistently improves. Therefore, we set the number of Transformer layers to 22 with 611 million trainable parameters as the optimal setting. Tab. 5 showcases the required resources for models of three different scales. Fig. 7 offers visual comparisons across models of varying scales, illustrating a clear trend: models with fewer parameters underperform on the HPSv2 benchmark, frequently resulting in distorted images and omitted details, yet they may still generate acceptable outcomes. Significantly, the visual quality diverges as model size increases, highlighting the potential for scaling up masked image modeling to enhance instruction-image alignment and elevate generation quality.

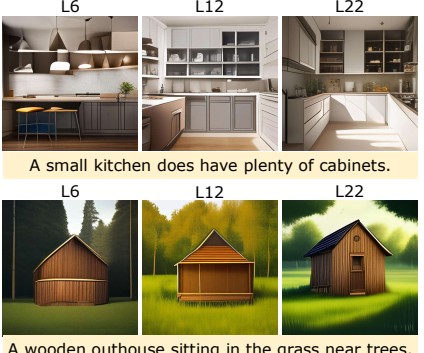

Figure 7: Examples generated by models at different scales: $1^{st}$ column for 6 layers, $2^{nd}$ column for 12 layers and $3^{rd}$ column for 22 layers.

## 5 Limitations and Social Impact

**Limitations.** Despite Fantasy achieving competitive performance in text-image alignment and visual appeal, it requires improvements in handling complex scenes. We propose two possible strategies to overcome the challenge in future research: Firstly, augmenting the dataset with high-quality images can enhance diversity and refine the model. Secondly, since the scale of the masked image generator affects instruction-image alignment, training an upscale image generator based on higher resolution left further explored.

**Social Impact.** Generative models for media bring both benefits and challenges. They foster creativity and make technology more accessible, yet pose risks by facilitating the creation of manipulated content, spreading misinformation, and exacerbating biases, particularly affecting women with deep fakes. Concerns also include the potential exposure of sensitive training data collected without consent. Despite generative models potentially offering better data representation, the impact of combining adversarial training with likelihood-based objectives on data distortion remains a crucial research area. Ethical considerations of these models are significant and require thorough exploration.

## 6 Conclusion

In this paper, we introduce Fantasy, a lightweight and efficient text-to-image model that combines Large Language Models (LLMs) with a transformer-based masked image modeling (MIM), effectively transferring semantic understanding capabilities from LLMs to the text-to-image generation. With our proposed two-stage training strategy and high-quality dataset, Fantasy significantly reduces computational requirements while producing high-fidelity images. Extensive experiments demonstrate that Fantasy achieves comparable performance to models trained with significantly more computational resources, illustrating the viability of our approach and suggesting potential efficient scalability to even larger masked image modeling for text-to-image generation.

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
