# OpenReview forum: "Fantasy: Transformer Meets Transformer in Text-to-Image Generation"
_NeurIPS.cc/2024/Conference — Submitted to NeurIPS 2024_

### Official Review · Reviewer_aiiy · 2024-06-14

**Soundness:** 3
**Presentation:** 3
**Contribution:** 2
**Rating:** 7
**Confidence:** 2

**Summary:**

The paper proposes Fantasy, a T2I model based fully on transformers (except for the VQGAN for the latents encoding and decoder):
* A __fine-tuned LLM__ (based on Phi-2) for the text encoding
* A image generator based on the MIM (Masked Image Modelling) approach

The training happens in two stages, a generic stage for aligning the generator the the frozen Phi-2 features, followed by a fine-tuning stage where the Phi-2 encoder is fine-tuned alongside the MIM transformer.

The results on human evaluations are convincing, putting Fantasy alongside models that require larger computational resources, while the FID results are less convincing (due to the image being smooth according to the authors).

**Strengths:**

Novelty:
* The LLM is __fine-tuned__ but only in the second stage of training, this approach is new and makes sense

Accessiblity:
* The 2 stage pre-training is already standard practice
* The Phi-2 model is available, it is likely that this approach works for other available models (Phi-3? It could be interesting to test)
* The model size allows the model to be trained in a reasonable time

**Weaknesses:**

Performance:
* The FID scores are not competitive and the authors describe why: the image are smooth => it seems that the human evaluations still rank Fantasy at the top on visual appeal, but it might be that if the question was "visual realism" they might prefer a different model
* Results are available for 256px, and a 600M parameters MIM generator, there is no proof that this method scales (we know that diffusion models based on UNet have trouble scaling for instance)

**Questions:**

n/a

---

> ### Author Rebuttal · Authors · 2024-08-06
>
> We appreciate for the valuable feedback and address the concerns as follows.
> ### W1: Explanation for Chosen Benchmarks.
> This is a good question. Several articles have noted that FID often **misaligns** with human evaluations, has limitations in assessing model quality, and is affected by factors such as image resolution and style. Moreover, since we use some generated data as a part of training source, there is a **domain gap** between Fantasy-generated images and the ground truth for FID-30K. To more thoroughly assess the images generated by Fantasy, we additionally choose HPSv2 as an evaluation metric (Table 1 in the paper), which aligns better with human evaluations.
>
>
> ### W2: Scaling Laws for Fantasy.
> We appreciate these concerns. Due to limitations in training data and computational resources, Fantasy’s training is currently limited to a 22-layer transformer-based MIM. Despite these constraints, Section 4.2 explores Fantasy’s scale-down performance, reducing from 22 layers to 6 layers. Additionally, Section 4.2 provides preliminary evidence that scaling laws validated for diffusion-based T2I models also apply to MIM-based T2I models. While diffusion models based on UNet face challenges with scaling, the scalability of Transformers has been well demonstrated in ViTs. Further exploration of scaling up the depth of MIM-based T2I models will be pursued in future work.

---

> ### Author Response · Authors · 2024-08-13
>
> Dear Reviewer aiiy,
>
> We sincerely thank you for your insightful and constructive feedback. We have incorporated your suggestions and responded comprehensively to in the comments.
>
> We highlight our **core contributions** and **distinguish our method from other two-stage training models** for all reviewers. We address the **Scaling Study’s significance and future directions**, and **clarify our benchmark selection** as recommended.
>
> We hope that our study will be well-received as a valuable contribution to the NeurIPS' focus on theory & application. We are available for any further discussions or inquiries the reviewers may have during the reviewer-author discussion period.
>
> Best regards,
>
> Authors

---

### Official Review · Reviewer_soBM · 2024-07-12

**Soundness:** 3
**Presentation:** 3
**Contribution:** 3
**Rating:** 5
**Confidence:** 4

**Summary:**

This paper proposes an efficient text-to-image generation model that integrates LLM and MIM. It demonstrates that MIM can achieve comparable performance. Unlike commonly used text encoders like CLIP and T5, this study introduces an efficient decoder-only LLM, phi-3, achieving better semantic understanding. The effectiveness of the method is validated through a newly proposed two-stage training approach and sufficient experiments.

**Strengths:**

1. The paper is well-written with clear logic.
2. The use of MIM and LLM for image generation introduces a novel approach.
3. The two-stage training method improves the generation results.

**Weaknesses:**

1. The quality of the generated images does not yet match that of existing methods (e.g., pixart-alpha, SDXL), with some loss of detail. This is noticeable from the comparison in column B of Figure 5.
2. Some aspects of the methodology could be clearer, and the overall coherence of the approach could be strengthened.
3. While the proposed method demonstrates efficiency advantages, particularly in faster training convergence, this can be influenced by various factors. However, the related experiments in the paper could be more comprehensive.
4. The semantic accuracy of the generated images, a potential strength of Fantasy, is not fully demonstrated in the paper. For instance, the model's ability to handle prompts with multiple entities, color attribute descriptions, or retaining key elements in long text inputs is not adequately showcased.

**Questions:**

1. The results in Figure 1 should be based on a consistent benchmark for image quality. Could you provide more detailed information about this benchmark?
2. Why is Phi-2 used for the LLM? If it is interchangeable, would it be possible to include comparison experiments using CLIP or T5?
3. Does the model also have advantages in inference speed, or is it comparable to existing methods?
4. The semantic accuracy of the generated images should be a strength of Fantasy, but this is not fully demonstrated in the paper. For example, it would be beneficial to show how the model handles prompts with multiple entities, color attribute descriptions, or retains key elements in long text inputs.

**Limitations:**

Yes.

---

> ### Author Rebuttal · Authors · 2024-08-06
>
> We appreciate the valuable feedback and address the concerns as follows.
>
> ### W1: Discussion about the generated images.
> There are currently many powerful diffusion-based T2I methods (e.g., pixart-$\alpha$, SDXL) that generate images with excellent visual appeal and details. However, our goal is not to develop a powerful MIM-based T2I method integrated with LLMs to produce highly detailed images that surpass diffusion models. Instead, we aim to create a lightweight T2I network with good text faithfulness by combining LLMs and MIM, and explore the limits of compressing such a T2I network.
>
> Due to the different methods of image generation, Fantsy’s visual appeal is limited by the image resolution. We constrain the image resolution to 256×256 pixels to ensure efficient computation, preventing the generation of highly detailed images. Our hierarchical training strategy prioritizes text understanding and alignment over visual detail, a deliberate trade-off to achieve efficient training with limited resources. preventing the generation of highly detailed images. The task of marrying LLM with MIM to produce images with rich detail and visual appeal will be left for future work.
>
>
> Additionally, Figure 4 in the attachment provides more examples of images generated by Fantasy, where we have highlighted the corresponding nouns and their attributes. These images demonstrate strong text faithfulness, proving that Fantasy, as a lightweight T2I network, is effective. Furthermore, there is potential for developing a larger-scale T2I network that combines MIM and LLM to generate richer and more text-aligned images in future.
>
> ### W2: Clearer Description of Fantasy
> We appreciate the careful review and will revise in the updated manuscript according to the feedback.
>
> ### W3: Details about Efficient Training
> We totally agree that faster training convergence can be influenced by various factors, such as pre-encoded data and training precision. Currently, we only test the training efficiency of different model precision and find that training speed is fastest with bf16 compared to fp16 and fp32. In the future, we will conduct more related experiments and include them in the updated manuscript.
>
> ### W4: Supplementary Information for Generated Images.
> We really appreciate this advice. In Figure 4 of the attachment, we provide more generated images with long text inputs. For better visualization, we highlight multiple entities and color attribute descriptions in the text inputs that appear in the generated images.
>
> ### Q1: Explanation of Figure 1.
> Thank you for pointing this out. We apologize for the confusion with the previous version and redraw Figure 1 as shown in the attachment. Figure 1 compares training costs and generation quality of different models. Circle sizes indicate image quality improvements. We categorize FID into three levels: FID < 7.5 as level one (Pixart-$\alpha$), 7.5 < FID < 15 as level two (ParaDiffusion, DALLE2, SDv1.5), and FID > 15 as level three (Fantasy, WÜRSTCHEN).
>
> ### Q2: Why use Phi-2?
> This is a good question. Phi-2 is a lightweight decoder-only LLM, it cannot be replaced by CLIP or T5, but can be optimized by using Phi-3 or other lightweight decoder-only LLMs in the future work.
>
> CLIP’s text encoder employs an absolute positional embedding limited to 77 tokens, and LongCLIP reveals that the actual effective text length for CLIP is only 20 tokens, which significantly limits CLIP’s text understanding capabilities. While encoder-decoder LLMs have been explored in various works, such as Muse and Pixart-$\alpha$, decoder-only LLMs have proven to perform better in text understanding tasks. As mentioned in Section 2.2.2, fine-tuning LLMs is crucial to leverage their enhanced semantic comprehension and generalization potential. Due to constraints in training data and computational resources, only a lightweight LLM like Phi-2 allows us to efficiently perform full fine-tuning. During the fine-tuning stage, we compare full fine-tuning with LoRA fine-tuning (Figure 3 in the attachment). The full fine-tuning approach results in lower loss in both training and evaluation compared to LoRA tuning. This is likely because direct LoRA tuning can diminish the capabilities of LLMs in the T2I process, and larger-scale LLMs like T5 do not support efficient full fine-tuning within our resource constraints.
>
>
> ### Q3: Inference speed of Fantasy.
> Fantasy has a significant inference speed advantage over diffusion-based models and is currently comparable to existing MIM-based methods. In specific, Fantasy requires 1.2 seconds to infer a single image with 32 sampling steps, similar to Muse’s inference speed of 1.3 seconds, and 2 times faster than Stable Diffusion v1.4. Due to the use of discrete tokens and parallel decoding, Fantasy is more efficient during inference. However, inference speed is influenced by various factors, including sampling steps and optimization of inference code. In the current work, we aim to optimize the memory requirements of Fantasy, leaving the optimization for accelerating inference speed for future work.
>
> ### Q4: Supplementary for Generated Images.
> We really appreciate this advice. In Figure 4 of the attachment, we provide more generated images with long text inputs. For better visualization, we highlight multiple entities and color attribute descriptions in the text inputs that appear in the generated images.

---

> ### Author Response · Authors · 2024-08-13
>
> Dear Reviewer soBM,
>
> We sincerely thank you for your insightful and constructive feedback. We have incorporated your suggestions and responded comprehensively to in the comments.
>
> We highlight our **core contributions** for all reviewers. We explain our **use of Phi-2**, **image detail limitations**. We also **describe the Figure 1** in the paper, and **compare the inference speed** with others. Following your advice, we **detail additional training steps** and include **more generated images** with long texts in the PDF.
>
> We hope that our study will be well-received as a valuable contribution to the NeurIPS' focus on theory & application. We are available for any further discussions or inquiries the reviewers may have during the reviewer-author discussion period.
>
> Best regards,
>
> Authors

---

> > ### Comment · Reviewer_soBM · 2024-08-13
> >
> > Thank you for providing the detailed answers. My concerns have been resolved, thus, I will increase the rating by 1. I recommend that the authors incorporate the changes discussed in the rebuttal into the final revision.

---

> > > ### Author Response · Authors · 2024-08-13
> > >
> > > Dear Reviewer soBM,
> > >
> > > Thank you very much for your feedback and for increasing the rating based on our rebuttal. We appreciate your suggestion to incorporate the changes discussed in the rebuttal into the final revision, and will ensure that these adjustments are clearly reflected in the final version of our paper.
> > >
> > > Thank you once again for your constructive critique and guidance throughout this review process.
> > >
> > > Best regards,
> > >
> > > Authors

---

### Official Review · Reviewer_AAcU · 2024-07-14

**Soundness:** 2
**Presentation:** 2
**Contribution:** 2
**Rating:** 4
**Confidence:** 4

**Summary:**

This paper proposes a technique for training transformer based masked image modeling in an efficient way. Two main contributions include (1) use of a LLM decoder as text embeddings, and (2) Two-stage training strategy for MIM models. Experimental results show good generation quality.

**Strengths:**

- The use of LLMs as text encoders seem interesting.
- Two-stage training approach makes sense. First, the use of pretraining data helps the model learn a general text-image model, and the high quality alignment data can improve the quality of generations.
- Training models on low resources seem appealing.

**Weaknesses:**

- I don't see anything new proposed in this paper. The authors simply use Phi-2 model as text encoder with MIM models, and use two-stage training.
- Even two-stage training is not something new to image synthesis. People have been doing aesthetic finetuning to improve image quality in diffusion models (eg. stable diffusion). The authors extend this to instruction-image data.
- The quality of generated images are not very impressive. When zoomed in, we notice a lot of visible artifacts. The generated images are also flat and doesn't have a lot of details.

**Questions:**

- Why use Phi-2 model when there are many LLMs available? Is this for efficiency?

---

> ### Author Rebuttal · Authors · 2024-08-06
>
> We appreciate the valuable feedback and address the concerns as follows.
>
> ### W1: Core Contributions of Fantasy.
> We propose a novel T2I framework by combining decoder-only LLM with transformer-based image generators to achieve the balance of effectiveness and efficiency. Our approach aims to empower smaller models with stronger generative capabilities. We describe the details of our designs in the author rebuttal and will revise in the updated manuscript.
>
> ### W2: Explanation for Two-stage Training.
>
> This is a good question. Our two-stage training is quite different from others used in image synthesis. The two-stage approach not only aims for improved aesthetics but also enhances text faithfulness by utilizing different data and training components in each stage. We are the first to fully fine-tune LLMs using only a small amount of high-quality real and synthetic data during T2I training. Thanks to our proposed hierarchical training strategy, we achieve state-of-the-art results among other open-source transformer-based models and rank above average compared to diffusion models (Table 1 in paper). For more details, please see the Q2 response of the rebuttal for all reviewers. We will include the details in our revised version.
>
>
> ### W3:  Why aren’t the generated images more impressive than those from diffusion models?
>
> While diffusion-based models are known for producing high-quality images with impressive detail and visual appeal, our focus with Fantasy is on developing a lightweight, efficient T2I network that integrates a lightweight decoder-only LLM effectively with a Transformer-based MIM for efficient training and long-form text alignment, rather than maximizing visual details. We achieve state-of-the-art results among other open-source transformer-based models and rank above average compared to diffusion models (Table 1 in the paper).
>
> Fantsy’s visual appeal is mainly limited by the image resolution. We constrain the image resolution to 256×256 pixels to ensure efficient computation, which inherently reduces details. Our hierarchical training strategy prioritizes text understanding and alignment over visual details, a deliberate trade-off to achieve efficient training with limited resources. Future work will focus on images with higher resolutions and appealing details.
>
> Figure 4 in the attachment provides more examples of images generated by Fantasy, where we have highlighted the corresponding nouns and their attributes. These images demonstrate semantic correctness and strong text faithfulness, proving that Fantasy, as a lightweight T2I network, is effective. We believe that our approach has significant potential for future enhancement.
>
> ### Q1: Why use Phi-2?
>
> We choose the lightweight Phi-2 for both efficiency and better image-text alignment. As mentioned in Section 2.2.2, fine-tuning LLMs is necessary to capitalize on their enhanced semantic comprehension and generalization potential. Due to constraints in training data and computational resources, only a lightweight LLM, Phi-2, allows us to efficiently perform full fine-tuning. We initially experiment with using LLaMA as the text encoder in the pre-training stage, but due to high costs and minimal loss reduction (Figure 2 in the attachment), we suspended this approach. Moreover, during the fine-tuning stage, we compare full fine-tuning with LoRA fine-tuning (Figure 3 in the attachment). The full fine-tuning approach results in lower loss in both training and evaluation compared to LoRA tuning. This is likely because direct LoRA tuning can diminish the capabilities of LLMs in the T2I process.

---

> ### Author Response · Authors · 2024-08-13
>
> Dear Reviewer AAcU,
>
> We sincerely thank you for your insightful and constructive feedback. We have incorporated your suggestions and responded comprehensively to in the comments.
>
> We highlight our **core contributions** and **distinguish our strategy from other hierarchical training strategy** both in data and components for all reviewers. We explain our **use of Phi-2** and **image detail limitations** as recommended.
>
> We hope that our study will be well-received as a valuable contribution to the NeurIPS' focus on theory & application. We are available for any further discussions or inquiries the reviewers may have during the reviewer-author discussion period.
>
> Best regards,
>
> Authors

---

> ### Author Response · Authors · 2024-08-14
>
> Dear Reviewer AAcU,
>
> As the discussion phase ends today, we will not be able to further clarify potential additional concerns. We would be very grateful if you could respond to our further comment and offer us an opportunity to address any questions you might have!
>
> Thank you again for your time and feedback!
>
> Best,
>
> Authors

---

### Official Review · Reviewer_5cW5 · 2024-07-27

**Soundness:** 2
**Presentation:** 2
**Contribution:** 2
**Rating:** 4
**Confidence:** 4

**Summary:**

To develop a resource-efficient, high-quality image generator for long instructions, the authors presented Fantasy, an efficient T2I generation model that integrates a lightweight decoder-only LLM and a transformer-based masked image modeling (MIM).

They demonstrate that with appropriate training strategies and high-quality data, MIM can also achieve comparable performance.

By incorporating pre-trained decoder-only LLMs as the text encoder, they observe a significant improvement in text fidelity compared to the widely used CLIP text encoder, enhancing the text image alignment.

Their training includes two stages: 1) large-scale concept alignment pre-training, and 2) fine-tuning with high-quality instruction-image data.

They conduct evaluation on FID, HPSv2 benchmarks, and human feedback, which demonstrate the competitive performance of Fantasy against other diffusion and autoregressive models.

**Strengths:**

- The author proposed a T2I framework that combines several more recent components and performed a series of comparisons, including both quantitative and human evaluations.

**Weaknesses:**

- the major concern of the work is unclear contributions. The claimed three contributions or core designs are quite similar with existing works.
- Efficient T2I network: there is no justification about why the network is “efficient”. Simply adopting a smaller LLM like Phi-2 can hardly be claimed as efficient network design.
- The hierarchical training strategy was also proposed before, it is not clear what is the difference with existing work.
- High quality data: the training data utilize Laion-2B and use existing filtering strategy. The collection high quality synthesized images from existing datasets.
- The evaluation metrics are mainly based on HPSv2, which has a limited range of values, e.g., HPSv2 has close values for SDv1.4 and SD2.0, e.g., 27.26 vs 27.48. Why SDXL is missing in Table 1?
- The author acknowledged that their model lags behind diffusion-based models in visual appeal, limited by the 8K size of VQGAN’s codebook and not targeting visual appeal. However, there is no solution or further study for solving this problem, which limits the scalability of the model.
- The scaling study in section 4.2 seems pretty premature and it is unclear what is the limit of the scaling. Increasing the model depth can improve the performance, which has been verified from previous work such as in https://arxiv.org/abs/2212.09748 or https://arxiv.org/abs/2404.02883.

**Questions:**

- what is the major difference with existing MIM based methods such as Muse, beside different components and data strategy?

**Limitations:**

I would encourage the authors to emphasize about the core contributions rather than combining everything together, which can hardly show significant performance improvement over existing public models.

---

> ### Author Rebuttal · Authors · 2024-08-06
>
> We appreciate the valuable feedback and address the concerns as follows.
> ### W1: Contributions Key Points.
> Our goal is to investigate whether combining LLM with transformer-based generators can enhance generative models by achieving a balance between effectiveness and efficiency. We also explore the scaling-down performance of our model which aims to empower smaller models with stronger generative capabilities. We emphasize the core designs as detailed in the author rebuttal and will revise in the updated manuscript.
> ### W2: Efficient T2I network.
> Thanks for highlighting this point. We aim to integrate an LLM with transformer-based MIM for T2I generation and use the lightweight Phi-2 given limited resource. Fantasy is efficient in both framework design and training strategy, optimizing computation and resource use.
> - **Structure.** Fantasy employ transformer-based generator and discrete tokens, which aligns with LLMs. By using the same structure for visual and text inputs, Fantasy enhances the effectiveness of the alignment between text and visual contents, so that smaller models can also achieve strong performance. Meanwhile, non-autoregressive Fantasy allows simultaneous computation of all positions during decoding, shortening dependency chains and speeding up image generation with better GPU utilization. More importantly, transformers’ scalable multi-layer structure enables efficient scaling, allowing for more compact image generators compared to diffusion models.
> - **Training.** By using MIM and fully fine-tuning the LLM, Fantasy increases the efficiency of data usage, allowing training with minimal high-quality data. Our hierarchical training strategy accelerates model fitting: the first stage achieves concept alignment, and the second mixes real and synthetic images with instructions to fine-tune the LLM for long text understanding. Compared to other models, Fantasy requires less training data and time (Figure 1 in the paper).
> ### W3: Difference of Hierarchical Training Strategy.
> While multi-stage training is common in T2I models, Fantasy’s strategy is unique in its components and data. We are the first to fully fine-tune LLMs with minimal high-quality real and synthetic data for T2I training. Our hierarchical strategy achieves state-of-the-art results among open-source transformer-based models and ranks above average compared to diffusion models (Table 1 in the paper). For more details, see the Q2 in the author rebuttal.
> ### W4: High-Quality Data.
> Our data comes from 3 sources: filtered Laion-2B with strategies like the CLIP score for pre-training, and higher-quality real and filtered synthesized data for the second stage. During fine-tuning, we use data from SAM-LLaVA without blurred human faces and synthetic images from JourneyDB and internal collections. To learn object relationships, we use NLTK to filter out noun phrase prompts and retain captions over 30 characters to enhance understanding.
> ### W5: HPSv2 Values for SDXL
> We agree that more baseline values are necessary. As shown in Table 1 of the attachment, we add HPSv2 values for SDXL, and Fantasy outperforms SDXL on nearly all metrics despite having fewer parameters and lower training costs. Note that for fair comparison, we set the image resolution to 512x512. We will include the result in the revised version.
> ### W6: Further Solutions for Scalability
> This is a good question. Although there is a gap in visual appeal compared to diffusion models, Fantasy effectively generates objects with their attributes and relationships. We freeze the VQGAN due to limited resources but plan to scale up by expanding its codebook when resources allow. Magvit-v2 shows that a larger vocabulary improves generative quality by allowing more diverse visual tokens for images and videos. The accuracy increasing as the codebook grows from 1K to 16K (ours is 8K). We will include these description in our revised version.
> ### W7: Meaning of Scaling Study
> We acknowledge that exploring scaling-up limits is restricted by training resource. Our experiments focus on finding the smallest effective scale for a T2I generation model, which we find to be 6 layers in our framework. This setup can still represent objects, attributes, and relationships in captions. As research advances, replacing Phi-2 with a more capable lightweight LLM and using higher-quality data could enhance Fantasy’s visual appeal and text faithfulness at this scale. Although limited to a 22-layer transformer, our findings suggest that scaling laws for diffusion-based T2I models also apply to MIM-based models. Further exploration of these scaling limits will be addressed in future work.
> ### Q1: Difference with MIM Methods
> The differences between Fantasy and the leading MIM-based T2I model, Muse, are as follows:
> - **Motivation.** Muse is the first to integrate MIM and LLM for T2I, while Fantasy aims at exploring a lightweight but effective T2I network.
> - **Model Components.** Muse includes a frozen encoder-decoder LLM, a trainable VQ, and a MIM generator, along with a MIM-based super-resolution module. Fantasy utilizes a frozen VQGAN, a tunable decoder-only LLM, and a lightweight transformer-based MIM.
> - **Data Strategy.** Muse is trained on Imagen with 460M text-image pairs in a single stage, and Fantasy is only trained on 16M text-image pairs with the mixture of real and synthetic data for two stage.
> - **Training Strategy.** Fantasy freeze the VQGAN and is the first to fully fine-tune LLM with a two-stage training strategy. Muse train the VQ and freeze the LLM, which is suboptimal for merely aligning text embeddings with visual features.
> - **Model Scale.** Fantasy targets a lightweight, efficient T2I network with models from 0.25B to 0.6B, while Muse offers larger models (0.6B and 3B) with examples only for the 3B model.

---

> ### Author Response · Authors · 2024-08-13
>
> Dear Reviewer 5cW5,
>
> We sincerely thank you for your insightful and constructive feedback. We have incorporated your suggestions and responded comprehensively to in the comments.
>
> We highlight our **core contributions** and **distinguish our strategy from other hierarchical training strategy** both in data and components for all reviewers. We refine our **efficiency definition**, **contrast our approach with MUSE**, and **incorporate baseline results** as recommended. We also address the **Scaling Study’s significance and future directions**.
>
> We hope that our study will be well-received as a valuable contribution to the NeurIPS' focus on theory & application. We are available for any further discussions or inquiries the reviewers may have during the reviewer-author discussion period.
>
> Best regards,
>
> Authors

---

> ### Author Response · Authors · 2024-08-14
>
> Dear Reviewer 5cW5,
>
> As the discussion phase ends today, we will not be able to further clarify potential additional concerns. We would be very grateful if you could respond to our further comment and offer us an opportunity to address any questions you might have!
>
> Thank you again for your time and feedback!
>
> Best,
>
> Authors

---

### Author Rebuttal · Authors · 2024-08-06

We appreciate all the reviewers for their valuable feedback and will address several frequently mentioned issues below.
### Q1: Core Contributions of Fantasy.
We would like to emphasize our core contributions again. Our goal is to investigate whether combining LLM with transformer-based generators can enhance generative models by achieving a balance between effectiveness and efficiency. Our approach aims to empower smaller models with stronger generative capabilities. Our major contributions can be summarized as follows:

- We present Fantasy, a novel lightweight framework that integrates a decoder-only LLM and a Transformer-based MIM for text-to-image synthesis, allowing for long-form text alignment and efficient training.
- We show that our two-stage training strategy is the first to fully fine-tune a LLM in text-to-image generation with high-quality mixed real and synthetic data, thereby enabling MIM to achieve comparable performance with a significantly reduced training cost in terms of model size and data usage.
- We provide comprehensive validation of the model’s efficacy based on automated metrics and human feedback for visual appeal and text faithfulness, and further investigate the minimum viable scale of the model.


### Q2: Explanation for Two-stage Training.

We propose a novel hierarchical training strategy different from others used in text-to-image generation. The two-stage approach not only aims for improved aesthetics but also enhances text faithfulness by utilizing different data and training components in each stage.

- **Training data.** High-quality data is crucial for training, especially when aligning image-text pairs. However, due to insufficient real-world data for Fantasy, we supplement with generated data primarily from MidJourney. The first stage aimed to perform general text-image concept alignment. In the second stage, our primary goal is to enable Fantasy to understand long text instructions. Therefore, we use filtered Laion-2B with re-captioned long prompts for soft alignment, ensuring text length exceeds 30 characters and excluding prompts of only noun phrases. We prioritize generated images and include higher-quality real images compared to the first stage to increase diversity and prevent domain shifts associated with relying solely on generated data.

- **Training components.** We are the first to fully fine-tune an LLM in text-to-image generation. Though it is common to perform general text-image concept alignment by training the generator and projection layer during the pre-training stage, we want to emphasize the fine-tuning stage of Fantasy. Most T2I methods freeze the text encoder; however, as mentioned in Section 2.2.2, fine-tuning LLMs is necessary to capitalize on their enhanced semantic comprehension and generalization potential. Existing methods, such as ParaDiffusion, which utilizes LLaMA2 and tunes with LoRA in hierarchical training, and Lavi-bridge, which integrates different LLMs and tunes with LoRA in a single stage, seem limited in fully utilizing LLMs. Due to computational resource constraints, only the lightweight decoder-only LLM, Phi-2, enables us to perform full fine-tuning. During the fine-tuning stage, we compare full fine-tuning with LoRA fine-tuning (Figure 3 in the attachment). The full fine-tuning approach shows better performance in both training and evaluation compared to LoRA tuning, which demonstrates that full fine-tuning can better leverage the strong text understanding ability of LLMs.

---

### Author Response · Authors · 2024-08-13

Dear Reviewers, Area Chairs, Senior Area Chairs, and Program Chairs,

We sincerely thank the Reviewers for their insightful and constructive feedback. We have incorporated their suggestions and responded comprehensively to their comments.

- We highlight our **core contributions** and distinguish our method from other two-stage training models for all reviewers.
- We address the **Scaling Study**’s significance and future directions with 5cW5 and aiiy, and explain our **use of Phi-2** and **image detail limitations** to AAcU and soBM.
- We also refine our **efficiency definition**, contrast our approach with MUSE, and incorporate baseline results as recommended by 5cW5.
- Following soBM’s advice, we detail additional training steps and include **more generated images** with long texts in the PDF.
- Our **benchmark selections** are clarified for aiiy.

We hope that our study will be well-received as a valuable contribution to the NeurIPS' focus on theory & application. We are available for any further discussions or inquiries the reviewers may have during the reviewer-author discussion period.

Best regards,

Authors

---

### Decision · Program_Chairs · 2024-09-25

**Decision:**

Reject

**Comment:**

The paper had 3 borderline reviews and 1 accept.

- All reviewers point out that the quality of the images is not up to par with current models and there are visible artifacts / smoothness.
- The reviewers also pointed out that they fail to see significant contributions in the paper: it combines masked image generators with a decoder LLM (the only part being different is the use of a decoder LLM).
- The reviewers also mention that the advantages of a decoder-only LLM aren't demonstrated in depth.

The AC agrees with the reviewers -- while fantasy introduces a lightweight training approach, there is a significant image quality gap which would have needed to be resolved for acceptance. The use of a decoder LLM is interesting, but lacks deeper analysis and ablations comparing against encoder LLMs to warrant acceptance.